# Collision Cross-entropy for Soft Class Labels and Entropy-based Clustering

## Abstract

We propose "collision cross-entropy" as a robust alternative to Shannon's cross-entropy (CE) loss when class labels are represented by soft categorical distributions y. In general, soft labels can naturally represent ambiguous targets in classification. They are particularly relevant for self-labeled clustering methods, where latent pseudo-labels $y$ are jointly estimated with the model parameters and uncertainty is prevalent. In case of soft labels $y$, Shannon's CE teaches the model predictions $\sigma$ to reproduce the uncertainty in each training example, which inhibits the model's ability to learn and generalize from these examples. As an alternative loss, we propose the negative log of "collision probability" that maximizes the chance of equality between two random variables, predicted class and unknown true class, whose distributions are $\sigma$ and $y$. We show that it has the properties of a generalized CE. The proposed collision CE agrees with Shannon's CE for one-hot labels $y$, but the training from soft labels differs. For example, unlike Shannon's CE, data points where $y$ is a uniform distribution have zero contribution to the training. Collision CE significantly improves classification supervised by soft uncertain targets. Unlike Shannon's, collision CE is symmetric for $y$ and $\sigma$, which is particularly relevant when both distributions are estimated in the context of self-labeled clustering. Focusing on discriminative deep clustering where self-labeling and entropy-based losses are dominant, we show that the use of collision CE improves the state-of-the-art. We also derive an efficient EM algorithm that significantly speeds up the pseudo-label estimation with collision CE.

## 1 Introduction and Motivation

Shannon's cross-entropy $H(y, \sigma)$ is the most common loss for training network predictions $\sigma$ from ground truth labels $y$ in the context of classification, semantic segmentation, etc. However, this loss may not be ideal for applications where the targets $y$ are soft distributions representing various forms of uncertainty. For example, this paper is focused on self-labeled classification [17, 1, 15, 16] where the ground truth is not available and the network training is done jointly with estimating latent *pseudo-labels* $y$. In this case soft $y$ can represent the distribution of label uncertainty. Similar uncertainty of class labels is also natural for supervised problems where the ground truth has errors [26, 41]. In any cases of label uncertainty, if soft distribution $y$ is used as a target in $H(y, \sigma)$, the network is trained to reproduce the uncertainty, see the dashed curves in Fig.1.

Our work is inspired by generalized entropy measures [33, 18]. Besides mathematical generality, the need for such measures *"stems from practical aspects when modelling real world phenomena though entropy optimization algorithms"* [30]. Similarly to $L_p$ norms, parametric families of generalized entropy measures offer a wide spectrum of options. The Shannon's entropy is just one of them. Other measures could be more "natual" for any given problem.

A simple experiment in Figure 2 shows that Shannon's cross-entropy produces deficient solutions for soft labels $y$ compared to the proposed *collision cross-entropy*. The limitation of the standard cross-entropy is that it encourages the distributions $\sigma$ and $y$ to be equal, see the dashed curves in Fig.1. For example, the model predictions $\sigma$ are trained to copy the uncertainty of the label distribution $y$, even when $y$ is an uninformative uniform distribution. In contrast, our collision cross-entropy (the solid curves) gradually weakens the training as $y$ gets less certain. This numerical property of our cross-entropy follows from its definition (9) - it maximizes the probability of "collision", which is an event when two random variables sampled from the distributions $\sigma$ and $y$ are equal. This means that the predicted class value is equal to the latent label. This is significantly different from the $\sigma = y$ encouraged by the Shannon's cross-entropy. For example, if $y$ is uniform then it does not matter what the model predicts as the probability of collision $\frac{1}{K}$ would not change.

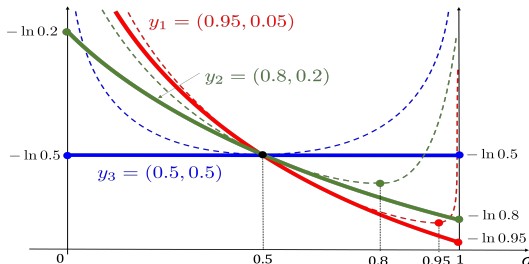

Figure 1: Collision cross-entropy $H_2(y, \sigma)$ in (9) for fixed soft labels $y$ (red, green, and blue). Assuming binary classification, all possible predictions $\sigma = (x, 1 - x) \in \Delta_2$ are represented by points $x \in [0, 1]$ on the horizontal axis. For comparison, thin dashed curves show Shannon's cross-entropy $H(y, \sigma)$ in (8). Note that $H$ converges to infinity at both endpoints of the interval. In contrast, $H_2$ is bounded for any non-hot $y$. Such boundedness suggests robustness to target errors represented by soft labels $y$. Also, collision cross-entropy $H_2$ gradually turns off the training (sets zero-gradients) as soft labels become highly uncertain (solid blue). In contrast, $H(y, \sigma)$ trains the network to copy this uncertainty, e.g. observe the optimum $\sigma$ for all dashed curves.

**Organization of the paper:** After the summary of our contributions below, Section 2 reviews the relevant background on self-labeling models/losses and generalized information measures for entropy, divergence, and cross-entropy. Then, Section 3 introduces our *collision cross entropy* measure, discusses its properties, related formulations of Rényi cross-entropy, and relation to noisy labels in fully-supervised settings. Section 4 formulates our self-labeling loss by replacing the Shannon's cross entropy term in a representative state-of-the-art formulation using soft pseudo-labels [16] with our collision-cross-entropy. The obtained loss function is convex w.r.t. pseudo-labels $y$, which makes estimation of $y$ amenable to generic projected gradient descent. However, Section 4 derives a much faster EM algorithm for estimating $y$. As common for self-labeling, optimization of the total loss w.r.t. network parameters is done via backpropagation. Section 5 presents our experiments, followed by conclusions.

**Summary of Contributions:** We propose the *collision cross-entropy* as an alternative to the standard Shannon's cross-entropy mainly in the context of self-labeled classification with soft pseudo-labels. The main practical advantage is its robustness to uncertainty in the labels, which could also be useful in other applications. The definition of our cross-entropy has an intuitive probabilistic interpretation that agrees with the numerical and empirical properties. Unlike the Shannon's cross-entropy, our formulation is symmetric w.r.t. predictions $\sigma$ and pseudo-labels $y$. This is a conceptual advantage since both $\sigma$ and $y$ are estimated/optimized distributions. Our cross-entropy allows efficient optimization of pseudo-labels by a proposed EM algorithm, that significantly accelerates a generic projected gradient descent. Our experiments show consistent improvement over multiple examples of unsupervised and semi-supervised clustering, and several standard network architectures.

## 2    Background Review

We study a new generalized cross-entropy measure in the context of deep clustering. The models are trained on unlabeled data, but applications with partially labeled data are also relevant. Self-labeled deep clustering is a popular area of research [5, 31]. More recently, the-state-of-the-art is achieved by discriminative clustering methods based on maximizing the mutual information between the input and the output of the deep model [3]. There is a large group of relevant methods [22, 10, 15, 17, 1, 16] and we review the most important loss functions, all of which use standard information-theoretic measures such as Shannon's entropy. In the second part of this section, we overview the necessary mathematical background on the generalized entropy measures, which are central to our work.

## 2.1 Information-based Self-labeled Clustering

The work of Bridle, Heading, and MacKay from 1991 [3] formulated *mutual information* (MI) loss for unsupervised discriminative training of neural networks using probability-type outputs, e.g. *softmax* $\sigma : \mathcal{R}^K \to \Delta^K$ mapping $K$ logits $l_k \in \mathcal{R}$ to a point in the probability simplex $\Delta^K$. Such output $\sigma = (\sigma_1, \ldots, \sigma_K)$ is often interpreted as a posterior over $K$ classes, where $\sigma_k = \frac{\exp l_k}{\sum_i \exp l_i}$ is a scalar prediction for each class $k$.

The unsupervised loss proposed in [3] trains the model predictions to keep as much information about the input as possible. They derived an estimate of MI as the difference between the average entropy of the output and the entropy of the average output

$$L_{mi} \quad := \quad -MI(c, X) \quad \approx \quad \overline{H(\sigma)} - H(\overline{\sigma}) \tag{1}$$

where $c$ is a random variable representing class prediction, $X$ represents the input, and the averaging is done over all input samples $\{X_i\}_{i=1}^M$, *i.e.* over $M$ training examples. The derivation in [3] assumes that softmax represents the distribution $\Pr(c|X)$. However, since softmax is not a true posterior, the right hand side in (1) can be seen only as an MI loss. In any case, (1) has a clear discriminative interpretation that stands on its own: $H(\overline{\sigma})$ encourages "fair" predictions with a balanced support of all categories across the whole training data set, while $\overline{H(\sigma)}$ encourages confident or "decisive" prediction at each data point implying that decision boundaries are away from the training examples [11]. Generally, we call clustering losses for softmax models "information-based" if they use measures from the information theory, e.g. entropy.

Discriminative clustering loss (1) can be applied to deep or shallow models. For clarity, this paper distinguishes parameters $\mathbf{w}$ of the *representation* layers of the network computing features $f_{\mathbf{w}}(X) \in \mathcal{R}^N$ for any input $X$ and the linear classifier parameters $\mathbf{v}$ of the output layer computing $K$-logit vector $\mathbf{v}^\top f$ for any feature $f \in \mathcal{R}^N$. The overall network model is defined as

$$\sigma(\mathbf{v}^\top f_{\mathbf{w}}(X)). \tag{2}$$

A special "shallow" case in (2) is a basic linear discriminator

$$\sigma(\mathbf{v}^\top X) \tag{3}$$

directly operating on low-level input features $f = X$. Optimization of the loss (1) for the shallow model (3) is done only over linear classifier parameters $\mathbf{v}$, but the deeper network model (2) is optimized over all network parameters $[\mathbf{v}, \mathbf{w}]$. Typically, this is done via gradient descent or backpropagation [35, 3].

Optimization of MI losses (1) during network training is mostly done with standard gradient descent or backpropagation [3, 22, 15]. However, due to the entropy term representing the decisiveness, such loss functions are

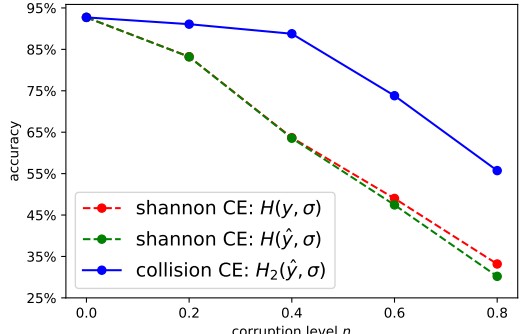

Figure 2: Robustness to label uncertainty: collision cross-entropy (9) vs Shannon's cross-entropy (8). The test uses ResNet-18 architecture on fully-supervised *Natural Scene* dataset [27] where we corrupted some labels. The horizontal axis shows the percentage $\eta$ of training images where the correct ground truth labels were replaced by a random label. Both losses trained the model using soft target distributions $\hat{y} = \eta * u + (1-\eta) * y$ representing the mixture of one-hot distribution $y$ for the observed corrupt label and the uniform distribution $u$, as recommended in [26]. The vertical axis shows the test accuracy. Training with the collision cross-entropy is robust to much higher levels of label uncertainty. As discussed in the last part of Sec.3, in the context of classification supervised by hard noisy labels, collision CE with soft labels can be related to the forward correction methods [28].

non-convex and present challenges to the gradient descent. This motivates alternative formulations and optimization approaches. For example, it is common to incorporate into the loss auxiliary variables $y$ representing *pseudo-labels* for unlabeled data points $X$ and to estimate them jointly with optimization of the network parameters [10, 1, 16]. Typically, such *self-labeling* approaches to unsupervised network training iterate optimization of the loss over pseudo-labels and network parameters, similarly to the Lloyd's algorithm for $K$-means [2]. While the network parameters are still optimized via gradient descent, the pseudo-labels can be optimized via more powerful algorithms.

For example, self-labeling in [1] uses the following constrained optimization problem with discrete pseudo-labels $y$

$$L_{ce} \;=\; \overline{H(y,\sigma)} \qquad s.t. \;\; y \in \Delta^K_{0,1} \;\; and \;\; \bar{y} = u \tag{4}$$

where $\Delta^K_{0,1}$ are *one-hot* distributions, *i.e.* corners of the probability simplex $\Delta^K$. Training the network predictions $\sigma$ is driven by the standard *cross entropy* loss $H(y,\sigma)$, which is convex assuming fixed (pseudo) labels $y$. With respect to variables $y$, the cross entropy is linear. Without the balancing constraint $\bar{y} = u$, the optimal $y$ corresponds to the hard $\arg\max(\sigma)$. However, the balancing constraint converts this into an integer programming problem that can be solved approximately via *optimal transport* [9]. The cross-entropy in (4) encourages the predictions $\sigma$ to approximate one-hot pseudo-labels $y$, which implies the decisiveness.

Self-labeling methods for unsupervised clustering can also use soft pseudo-labels $y \in \Delta^K$ as target distributions in cross-entropy $H(y,\sigma)$. In general, soft targets $y$ are common in $H(y,\sigma)$, e.g. in the context of noisy labels [41, 38]. Softened targets $y$ can also assist network calibration [12, 26] and improve generalization by reducing over-confidence [29]. In the context of unsupervised clustering, cross-entropy $H(y,\sigma)$ with soft pseudo-labels $y$ approximates the decisiveness since it encourages $\sigma \approx y$ implying $H(y,\sigma) \approx H(y) \approx H(\sigma)$ where the latter is the first term in (1). Instead of the hard constraint $\bar{y} = u$ used in (4), the soft fairness constraint can be represented by KL divergence $KL(\bar{y} \,\|\, u)$, as in [10, 16]. In particular, [16] formulates the following self-labeled clustering loss

$$L_{ce+kl} \;=\; \overline{H(y,\sigma)} \;+\; KL(\bar{y} \,\|\, u) \tag{5}$$

encouraging decisiveness and fairness as discussed. Similarly to (4), the network parameters in loss (5) are trained by the standard cross-entropy term, but optimization over relaxed pseudo-labels $y \in \Delta^K$ is relatively easy due to convexity. While there is no closed-form solution, the authors offer an efficient approximate solver for $y$. Iterating steps that estimate pseudo-labels $y$ and optimize the model parameters resembles the Lloyd's algorithm for K-means. The results in [16] also establish a formal relation between the loss (5) and the $K$-means objective.

## 2.2  Generalized Entropy Measures

Below, we review relevant generalized formulations of the information-theoretic concepts: entropy, divergence, and cross-entropy. Rényi [33] introduced the *entropy of order* $\alpha > 0$ for any probability distribution $p$

$$H_\alpha(p) \;:=\; \frac{1}{1-\alpha} \ln \sum_k p_k^\alpha \qquad (\alpha \neq 1)$$

derived as the most general measure of uncertainty in $p$ satisfying four intuitively evident postulates. The entropy measures the average information and the order parameter $\alpha$ relates to the power of the corresponding mean statistic [44]. The general formula above includes the Shannon's entropy

$$H(p) \;=\; -\sum_k p_k \ln p_k$$

as a special case when $\alpha \to 1$. The quadratic or second-order Rényi entropy

$$H_2(p) \;:=\; -\ln \sum_k p_k^2 \tag{6}$$

is also known as a *collision entropy* since it is a negative log-likelihood of a "collision" or "rolling double" when two i.i.d. samples from distribution $p$ have equal values.

Basic characterization postulates in [33] also lead to the general Rényi formulation of the *divergence*, also known as the *relative entropy*, of order $\alpha > 0$

$$D_\alpha(p \,|\, q) \;:=\; \frac{1}{\alpha - 1} \ln \sum_k p_k^\alpha \, q_k^{1-\alpha} \qquad (\alpha \neq 1)$$

defined for any pair of distributions $p$ and $q$. This reduces to the standard KL divergence when $\alpha \to 1$

$$D(p,q) = \sum_k p_k \ln \frac{p_k}{q_k} \tag{7}$$

169 and to the *Bhattacharyya distance* for $\alpha = \frac{1}{2}$.

170 Optimization of entropy and divergence [24] is fundamental to many machine learning problems
171 [37, 20, 19, 30], including pattern classification and cluster analysis [36]. However, the entropy-
172 related terminology is often mixed-up. For example, when discussing the *cross-entropy minimization*
173 *principle* (MinxEnt), many of the references cited earlier in this paragraph define *cross-entropy* using
174 the expression for KL-divergence (7). Nowadays, it is standard to define the Shannon's cross-entropy
175 as

$$H(p, q) \ = \ - \sum_k p_k \ln q_k. \tag{8}$$

176 One simple explanation for the confusion is that KL-divergence $D(p, q)$ and cross-entropy $H(p, q)$
177 as functions of $q$ only differ by a constant if $p$ is a fixed known target, which is often the case.

## 3   Collision Cross-Entropy

Minimizing divergence enforces proximity between two distributions, which may work as a loss
for training model predictions $\sigma$ with labels $y$, for example, if $y$ are ground truth one-hot labels.
However, if $y$ are pseudo-labels that are estimated jointly with $\sigma$, proximity between $y$ and $\sigma$ is not a
good criterion for the loss. For example, highly uncertain model predictions $\sigma$ in combination with
uniformly distributed pseudo-labels $y$ correspond to the optimal zero divergence, but this is not a very
useful result for self-labeling. Instead, all existing self-labeling losses for deep clustering minimize
Shannon's cross-entropy (8) that reduces the divergence and uncertainty at the same time

$$H(y, \sigma) \ \equiv \ D(y, \sigma) + H(y).$$

179 The entropy term corresponds to the "decisiveness" constraint in unsupervised discriminative clus-
180 tering [3, 17, 1, 15, 16]. In general, it is recommended as a regularizer for unsupervised and
181 semi-supervised network training [11] to encourage decision boundaries away from the data points
182 implicitly increasing the decision margins.

183 We propose a new form of cross-entropy

$$H_2(p, q) \ := \ - \ln \sum_k p_k \, q_k \tag{9}$$

184 that we call *collision cross-entropy* since it extends the collision entropy in (6). Indeed, (9) is the
185 negative log-probability of an event that two random variables with (different) distributions $p$ and $q$
186 are equal. When training softmax $\sigma$ with pseudo-label distribution $y$, the collision event is the exact
187 equality of the predicted class and the pseudo-label, where these are interpreted as specific outcomes
188 for random variables with distributions $\sigma$ and $y$. Note that the collision event, i.e. the equality of
189 two random variables, has very little to do with the equality of distributions $\sigma = y$. The collision
190 may happen when $\sigma \neq y$, as long as $\sigma \cdot y > 0$. Vice versa, this event is not guaranteed even when
191 $\sigma = y$. It will happen *almost surely* only if the two distributions are the same one-hot. However, if
192 the distributions are both uniform, the collision probability is only $1/K$.

As easy to check, the collision cross-entropy (9) can be equivalently represented as

$$H_2(p, q) \ \equiv \ - \ln cos(p, q) \ + \ \frac{H_2(p) + H_2(q)}{2}$$

193 where $cos(p, q)$ is the cosine of the angle between $p$ and $q$ as vectors in $\mathcal{R}^K$ and $H_2$ is the collision
194 entropy (6). The first term corresponds to a "distance" between the two distributions: it is non-
195 negative, equals 0 iff $p = q$, and $- \ln cos(\cdot)$ is a convex function of an angle, which can be interpreted
196 as a spherical metric. Thus, analogously to the Shannon's cross-entropy, $H_2$ is the sum of divergence
197 and entropy.

198 The formula (9) can be found as a definition of quadratic Rényi cross-entropy [30, 32, 46]. However,
199 we could not identify information-theoretic axioms characterizing a generalized cross-entropy. Rényi
200 himself did not discuss the concept of cross-entropy in his seminal work [33]. Also, two different
201 formulations of "natural" and "shifted" Rényi cross-entropy of arbitrary order could be found in
202 [44, 42]. In particular, the shifted version of order 2 agrees with our formulation of collision cross-
203 entropy (9). However, lack of postulates or characterization for the cross-entropy, and the existence of
204 multiple non-equivalent formulations did not give us the confidence to use the name Rényi. Instead,

we use "collision" due to its clear intuitive interpretation of the loss (9). But, the term "cross-entropy" is used only informally.

The numerical and empirical properties of the collision cross-entropy (9) are sufficiently different from the Shannons cross-entropy (8). Figure 1 illustrates $H_2(y,\sigma)$ as a function of $\sigma$ for different label distributions $y$. For confident $y$ it behaves the same way as the standard cross entropy $H(y,\sigma)$, but softer low-confident labels $y$ naturally have little influence on the training. In contrast, the standard cross entropy encourages prediction $\sigma$ to be the exact copy of uncertainty in distribution $y$. Self-labeling methods based on $H(y,\sigma)$ often "prune out" uncertain pseudo-labels [4]. Collision cross-entropy $H_2(y,\sigma)$ makes such heuristics redundant. We also demonstrate the "robustness to label uncertainty" on an example where the ground truth labels are corrupted by noise, see Fig.2. This artificial fully-supervised test is used only to compare the robustness of (9) and (8) in complete isolation from other terms in the self-labeled clustering losses, which are the focus of this work.

Due to the symmetry of the arguments in (9), such robustness of $H_2(y,\sigma)$ also works the other way around. Indeed, self-labeling losses are often used for both training $\sigma$ and estimating $y$: the loss is iteratively optimized over predictions $\sigma$ (i.e. model parameters responsible for it) and over pseudo-label distribution $y$. Thus, it helps if $y$ also demonstrates "robustness to prediction uncertainty".

**Soft labels vs noisy labels:** Our collision CE for soft labels, represented by distributions $y$, can be related to loss functions used for supervised classification with *noisy labels* [40, 28, 38], which assume some observed hard target labels $l$ that may not be true due to corruption or "noise". Instead of our probability of collision

$$\Pr(C = T) = \sum_k \Pr(C = k, T = k) = \sum_k \sigma_k y_k \equiv y^\top \sigma$$

between the predicted class $C$ and unknown true class $T$, whose distributions are prediction $\sigma$ and soft target $y$, they maximize the probability that a random variable $L$ representing a corrupted target equals the observed value $l$

$$\Pr(L = l) = \sum_k \Pr(L = l | T = k) \Pr(T = k) \approx \sum_k \Pr(L = l | T = k)\, \sigma^k \equiv Q_l\,\sigma$$

where the approximation uses the model predictions $\sigma^k$ instead of true class probabilities $\Pr(T = k)$, which is a significant assumption. Vector $Q_l$ is the $l$-th row of the *transition matrix* $Q$, such that $Q_{lk} = \Pr(L = l | T = k)$, that has to be obtained in addition to hard noisy labels $l$.

Our approach maximizing the collision probability based on soft labels $y$ is a generalization of the methods for hard noisy labels. Their transitional matrix $Q$ can be interpreted as an operator for converting any hard label $l$ into a soft label $y = Q^\top \mathbf{1}_l = Q_l$. Then, the two methods are numerically equivalent, though our statistical motivation is significantly different. Moreover, our approach is more general since it applies to a wider set of problems where the class target $T$ can be directly specified by a distribution, a soft label $y$, representing the target uncertainty. For example, in fully supervised classification or segmentation the human annotator can directly indicate uncertainty (odds) for classes present in the image or at a specific pixel. In fact, class ambiguity is common in many data sets, though for efficiency, the annotators are typically forced to provide one hard label. Moreover, in the context of self-supervised clustering, it is natural to estimate pseudo-labels as soft distributions $y$. Such methods directly benefit from our collision CE, as this paper shows.

## 4 Our Self-labeling Loss and EM

Based on prior work (5), we replace the standard cross-entropy with our collision cross-entropy to formulate our self-labeling loss as follows:

$$L_{CCE} \;:=\; \overline{H_2(y,\sigma)} + \lambda\, KL(\bar{y}\|u) \tag{10}$$

To optimize such loss, we iterate between two alternating steps for $\sigma$ and $y$. For $\sigma$, we use the standard stochastic gradient descent algorithms[34]. For $y$, we use the projected gradient descent (PGD) [7]. However, the speed of PGD is slow as shown in Table 1 especially when there are more classes. This motivates us to find more efficient algorithms for optimizing $y$. To derive such an algorithm, we made a minor change to (10) by switching the order of variables in the divergence term:

$$L_{CCE+} \;:=\; \overline{H_2(y,\sigma)} + \lambda\, KL(u\|\bar{y}) \tag{11}$$

Such change allows us to use the Jensen's inequality on the divergence term to derive an efficient EM algorithm while the quality of the self-labeled classification results is almost the same as shown in the Appendix D.

**EM algorithm for optimizing** $y$  We derive the EM algorithm introducing latent variables, $K$ distributions $S^k \in \Delta^M$ representing normalized support for each cluster over $M$ data points. We refer to each vector $S^k$ as a *normalized cluster* $k$. Note the difference with distributions represented by pseudo-labels $y \in \Delta^K$ showing support for each class at a given data point. Since we explicitly use individual data points below, we will start to carefully index them by $i \in \{1, \ldots, M\}$. Thus, we will use $y_i \in \Delta^K$ and $\sigma_i \in \Delta^K$. Individual components of distribution $S^k \in \Delta^M$ corresponding to data point $i$ will be denoted by scalar $S_i^k$.

First, we expand (11) introducing the latent variables $S^k \in \Delta^M$

$$L_{CCE+} \quad \overset{c}{=} \quad \overline{H_2(y,\sigma)} + \lambda\, H(u,\bar{y}) \tag{12}$$

$$= \quad \overline{H_2(y,\sigma)} - \lambda \sum_k u^k \ln \sum_i S_i^k \frac{y_i^k}{S_i^k M} \leq \quad \overline{H_2(y,\sigma)} - \lambda \sum_k \sum_i u^k S_i^k \ln \frac{y_i^k}{S_i^k M} \tag{13}$$

Due to the convexity of negative $\log$, we apply the Jensen's inequality to derive an upper bound, i.e. (13), to $L_{CCE+}$. Such bound becomes tight when:

$$\textbf{E step} : \qquad\qquad S_i^k = \frac{y_i^k}{\sum_j y_j^k} \tag{14}$$

Next, we derive the M step. Introducing the hidden variable $S$ breaks the fairness term into the sum of independent terms for pseudo-labels $y_i \in \Delta_K$ at each data point $i$. The solution for $S$ does not change (E step). Lets focus on the loss with respect to $y$. The collision cross-entropy (CCE) also breaks into the sum of independent parts for each $y_i$. For simplicity, we will drop all indices $i$ in variables $y_i^k$, $S_i^k$, $\sigma_i^k$. Then, the combination of CCE loss with the corresponding part of the fairness constraint can be written for each $y = \{y_k\} \in \Delta_K$ as

$$-\ln \sum_k \sigma_k y_k \quad - \quad \lambda \sum_k u_k S_k \ln y_k. \tag{15}$$

| | running time in sec. per iteration | | | number of iterations (to convergence) | | | running time in sec. (to convergence) | | |
|---|---|---|---|---|---|---|---|---|---|
| K | 2 | 20 | 200 | 2 | 20 | 200 | 2 | 20 | 200 |
| PGD ($\eta_1$) | $7.8e^{-4}$ | $2.9e^{-3}$ | $6.7e^{-2}$ | 326 | 742 | 540 | 0.25 | 2.20 | 36.25 |
| PGD ($\eta_2$) | $9.3e^{-4}$ | $3.3e^{-3}$ | $6.8e^{-2}$ | 101 | 468 | 344 | 0.09 | 1.55 | 23.35 |
| PGD ($\eta_3$) | $9.9e^{-4}$ | $3.2e^{-3}$ | $7.0e^{-2}$ | 24 | 202 | 180 | 0.02 | 0.65 | 12.60 |
| our EM | $1.8e^{-3}$ | $1.6e^{-3}$ | $5.1e^{-3}$ | 25 | 53 | 71 | 0.04 | 0.09 | 0.36 |

Table 1: Comparison of our EM algorithm to Projected Gradient Descent (PGD). $\eta$ is the step size. For $K = 2$, $\eta_1 \sim \eta_3$ are 1, 10 and 20 respectively. For $K = 20$ and $K = 200$, $\eta_1 \sim \eta_3$ are 0.1, 1 and 5 respectively. Higher step size leads to divergence of PGD.

First, observe that this loss must achieve its global optimum in the interior of the simplex if $S_k > 0$ and $u_k > 0$ for all $k$. Indeed, the second term enforces the "log-barier" at the boundary of the simplex. Thus, we do not need to worry about KKT conditions in this case. Note that $S_k$ might be zero, in which case we need to consider the full KKT conditions. However, the Property 1 that will be mentioned later eliminates such concern if we use positive initialization. For completeness, we also give the detailed derivation for such case and it can be found in the Appendix B.

Adding the Lagrange multiplier $\gamma$ for the simplex constraint, we get an unconstrained loss

$$-\ln \sum_k \sigma_k y_k \quad - \quad \lambda \sum_k u_k S_k \ln y_k \quad + \quad \gamma \left( \sum_k y_k - 1 \right)$$

that must have a stationary point inside the simplex. The following theorem indicates the way to solve the problem above. All the missing proofs can be found in Appendix A.

**Theorem 1. [M-step solution]:** *The sum $\sum_k y_k$ as in (16) is positive, continuous, convex, and monotonically decreasing function of $x$ on the specified interval. Moreover, there exists a unique solution $\{y_k\} \in \Delta_k$ and $x$ such that*

$$\sum_k y_k \quad \equiv \quad \sum_k \frac{\lambda u_k S_k}{\lambda u^\top S + 1 - \frac{\sigma_k}{x}} \quad = \quad 1 \quad and \quad x \in \left( \frac{\sigma_{max}}{1 + \lambda u^\top S}, \, \sigma_{max} \right] \tag{16}$$

The monotonicity and convexity of $\sum_k y_k$ with respect to $x$ suggest that the problem (16) formulated in Theorem 1 allows efficient algorithms for finding the corresponding unique solution. For example, one can use the iterative Newton's updates to search for $x$ in the specified interval. The following Lemma gives us a proper starting point

**Lemma 1.** *Assuming $u_k S_k$ is positive for each $k$, then the reachable left end point in Theorem 1 can be written as*

$$l := \max_k \frac{\sigma_k}{1 + \lambda u^\top S - \lambda u_k S_k}.$$

for Newton's method. The algorithm for M-step solution is summarized in Algorithm 1 in Appendix C. Note that we present the algorithm for only one data point, and we can easily and efficiently scale up for more data in a batch by using the Numba compiler. In the following, we give the property about the positivity of the solution. This property implies that if our EM algorithm has only (strictly) positive variables $S_k$ or $y_k$ at initialization, these variables will remain positive during all iterations.

**Property 1.** For any category $k$ such that $u_k > 0$, the set of strictly positive variables $y_k$ or $S_k$ can only grow during iterations of our EM algorithm for the loss (15) based on the collision cross-entropy.

Note that Property 1 does not rule out the possibility that $y_k$ may become arbitrarily close to zero during EM iterations. Empirically, we did not observe any numerical issues. The complete algorithm is given in Appendix C. Inspired by [39, 15], we also update our $y$ in each batch. Intuitively, updating $y$ on the fly can prevent the network from being easily trapped in some local minima created by the incorrect pseudo-labels.

## 5 Experiments

We apply our new loss to self-labeled classification problems in both shallow and deep settings, as well as semi-supervised modes. All the results are reproduced using either public codes or our own implementation under the same experimental settings for fair comparison. Our approach consistently achieves either the best or highly competitive results across all the datasets and is therefore more robust. All the missing details in the experiments can be found in Appendix E.

**Dataset**   We use four standard datasets: MNIST [25], CIFAR10/100 [43] and STL10 [8]. The training and test data are the same unless otherwise specified.

**Evaluation**   As for the evaluation of self-labeled classification, we set the number of clusters to the number of ground-truth categories. To calculate the accuracy, we use the standard Hungarian algorithm [23] to find the best one-to-one mapping between clusters and labels. We don't need this matching step if we use other metrics, i.e. NMI, ARI.

### 5.1 Clustering with Fixed Features

In this section, we test our loss as a proper clustering loss and compare it to the widely used Kmeans (generative) and other closely related losses (entropy-based and discriminative). We use the pretrained (ImageNet) Resnet-50 [14] to extract the features. For Kmeans, the model is parameterized by K cluster centers. Comparably, we use a one-layer linear classifier followed by softmax for all other losses including ours. Kmeans results were obtained using scikit-learn package in Python. To optimize the model parameters for other losses, we use

|  | STL10 | CIFAR10 | CIFAR100-20 | MNIST |
|---|---|---|---|---|
| Kmeans | 85.20%(5.9) | 67.78%(4.6) | 42.99%(1.3) | 47.62%(2.1) |
| MIGD [22] | 89.56%(6.4) | 72.32%(5.8) | 43.59%(1.1) | 52.92%(3.0) |
| SeLa [1] | 90.33%(4.8) | 63.31%(3.7) | 40.74%(1.1) | 52.38%(5.2) |
| MIADM [16] | 88.64%(7.1) | 60.57%(3.3) | 41.2%(1.4) | 50.61%(1.3) |
| Our | **92.33%(6.4)** | **73.51%(6.3)** | **43.72%(1.1)** | **58.4%(3.2)** |

Table 2: Comparison of different methods on clustering with fixed features extracted from Resnet-50. The numbers are the average accuracy and the standard deviation over trials. We use the 20 coarse categories for CIFAR100 similarly to others.

stochastic gradient descent. Here we report the average accuracy and standard deviation over 6 randomly initialized trials in Table 2.

## 5.2 Deep Clustering

In this section, we train a deep network to jointly learn the features and cluster the data. We test our method on both a small architecture (VGG4) and a large one (ResNet-18). The only extra standard technique we add here is self-augmentation following [15, 1, 6].

To train the VGG4, we use random initialization for network parameters. From Table 3, it can be seen that our approach consistently achieves the most competitive results in terms of accuracy (ACC). Most of the methods we compared in our work (including our method) are general concepts applicable to single-stage end-to-end training. To be fair, we tested all of them on the same simple architecture. But, these general methods can be easily integrated into other more complex systems with larger architecture such as ResNet-18.

In Table 4, we show the results using the pretext-trained network from SCAN [45] as initialization for our clustering loss as well as IMSAT and MIADM. We use only the clustering loss together with the self-augmentation (one augmentation per image). As shown in the table below, our method reaches a higher number with more robustness almost for every metric on all datasets compared to the SOTA method SCAN. More importantly, we consis-

|  | STL10 | CIFAR10 | CIFAR100-20 | MNIST |
|---|---|---|---|---|
| IMSAT [15] | 25.28%(0.5) | 21.4%(0.5) | 14.39%(0.7) | 92.90%(6.3) |
| IIC [17] | 24.12%(1.7) | 21.3%(1.4) | 12.58%(0.6) | 82.51%(2.3) |
| SeLa [1] | 23.99%(0.9) | 24.16%(1.5) | **15.34%(0.3)** | 52.86%(1.9) |
| MIADM [16] | 23.37%(0.9) | 23.26%(0.6) | 14.02%(0.5) | 78.88%(3.3) |
| Our | **25.98%(1.1)** | **24.26%(0.8)** | 15.14%(0.5) | **95.11%(4.3)** |

Table 3: Quantitative comparison of discriminative clustering-based classification methods with simultaneous feature training from the scratch. The network architecture is VGG-4. We reuse the code published by [17, 1, 15] and use our improved implementation of [16] (also for other tables).

| | CIFAR10 | | | CIFAR100-20 | | | STL10 | | |
|---|---|---|---|---|---|---|---|---|---|
| | ACC | NMI | ARI | ACC | NMI | ARI | ACC | NMI | ARI |
| SCAN [45] | 81.8% (0.3) | 71.2% (0.4) | 66.5% (0.4) | 42.2% (3.0) | **44.1%** (1.0) | 26.7% (1.3) | 75.5% (2.0) | 65.4% (1.2) | 59.0% (1.6) |
| IMSAT [15] | 77.64% (1.3) | 71.05% (0.4) | 64.85% (0.3) | 43.68% (0.4) | 42.92% (0.2) | 26.47% (0.1) | 70.23% (2.0) | 62.22% (1.2) | 53.54% (1.1) |
| MIADM [16] | 74.76% (0.3) | 69.17% (0.2) | 62.51% (0.2) | 43.47% (0.5) | 42.85% (0.4) | 27.78% (0.4) | 67.84% (0.2) | 60.33% (0.5) | 51.67% (0.6) |
| Our | **83.27%** (0.2) | **71.95%** (0.2) | **68.15%** (0.1) | **47.01%** (0.2) | 43.28% (0.1) | **29.11%** (0.1) | **78.12%** (0.1) | **68.11%** (0.3) | **62.34%** (0.3) |

Table 4: Quantitative comparison using network ResNet-18. The most related work MIADM (5) is also highlighted in all tables.

tently improve over the most related method, MIADM, by a large margin, which clearly demonstrates the effectiveness of our proposed loss together with the optimization algorithm.

## 5.3 Semi-supervised Classification

Although our paper is focused on self-labeled classification, we find it also interesting and natural to test our loss under semi-supervised settings where partial data is provided with ground-truth labels. We use the standard cross-entropy loss for labeled data and directly add it to the self-labeled loss to train the network initialized by the pretext-trained network following [45].

## 6 Conclusion

We propose a new collision cross-entropy loss. Such loss is naturally interpreted as measuring the probability of the equality between two random variables represented by the two distributions $\sigma$ and $y$, which perfectly fits the goal of self-labeled classification. It is symmetric w.r.t. the two distributions instead of treating one as the target, like the standard cross-entropy.

| | 0.1 | | 0.05 | | 0.01 | |
|---|---|---|---|---|---|---|
| | STL10 | CIFAR10 | STL10 | CIFAR10 | STL10 | CIFAR10 |
| Only seeds | 78.4% | 81.2% | 74.1% | 76.8% | 68.8% | 71.8% |
| + IMSAT [15] | 88.1% | 91.5% | 81.1% | 85.2% | 74.1% | 80.2% |
| + IIC [17] | 85.2% | 90.3% | 78.2% | 84.8% | 72.5% | 80.5% |
| + SeLa [1] | 86.2% | 88.6% | 79.5% | 82.7% | 69.9% | 79.1% |
| + MIADM [16] | 84.9% | 86.1% | 77.9% | 80.1% | 69.6% | 77.5% |
| + Our | **88.9%** | **92.3%** | **82.9%** | **86.2%** | **75.7%** | **82.4%** |

Table 5: Quantitative results for semi-supervised classification on STL10 and CIFAR10 using ResNet18. The numbers 0.1, 0.05 and 0.01 correspond to different ratio of labels used for supervision. "Only seeds" means we only use standard cross-entropy loss on seeds for training.

While the latter makes the network copy the uncertainty in estimated pseudo-labels, our cross-entropy naturally weakens the training on data points where pseudo labels are more uncertain. This makes our cross-entropy robust to labeling errors. In fact, the robustness works both for prediction and for pseudo-labels due to the symmetry. We also developed an efficient EM algorithm for optimizing the pseudo-labels. Such EM algorithm takes much less time compared to the standard projected gradient descent. Experimental results show that our method consistently produces top or near-top results on all tested clustering and semi-supervised benchmarks.

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

## A  Missing proofs

**Theorem 2. [M-step solution]:** *The sum $\sum_k y_k$ as in (17) is positive, continuous, convex, and monotonically decreasing function of $x$ on the specified interval. Moreover, there exists a unique solution $\{y_k\} \in \Delta_k$ and $x$ such that*

$$\sum_k y_k \quad \equiv \quad \sum_k \frac{\lambda u_k S_k}{\lambda u^\top S + 1 - \frac{\sigma_k}{x}} \quad = \quad 1 \quad and \quad x \in \left( \frac{\sigma_{max}}{1 + \lambda u^\top S} \, , \, \sigma_{max} \right] \quad (17)$$

*Proof.* All $y_k$ in (17) are positive, continuous, convex, and monotonically decreasing functions of $x$ on the specified interval. Thus, $\sum y_k$ behaves similarly. Assuming that $max$ is the index of prediction $\sigma_{max}$, we have $y_{max} \to +\infty$ when approaching the interval's left endpoint $x \to \frac{\sigma_{max}}{1 + \lambda u^\top S}$. Thus, $\sum y_k > 1$ for smaller values of $x$. At the right endpoint $x = \sigma_{max}$ we have $y_k \leq \frac{\lambda u_k S_k}{\lambda u^\top S}$ for all $k$ implying $\sum y_k \leq 1$. Monotonicity and continuity of $\sum y_k$ w.r.t. $x$ imply the theorem. $\qquad \square$

**Lemma 2.** *Assuming $u_k S_k$ is positive for each $k$, then the reachable left end point in Theorem 1 can be written as*

$$l := \max_k \frac{\sigma_k}{1 + \lambda u^\top S - \lambda u_k S_k}.$$

*Proof.* Firstly, we prove that $l$ is (strictly) inside the interior of the interval in Theorem 1. For the left end point, we have

$$l := \max_k \frac{\sigma_k}{1 + \lambda u^\top S - \lambda u_k S_k}$$
$$\geq \quad \frac{\sigma_{max}}{1 + \lambda u^\top S - \lambda u_{max} S_{max}}$$
$$> \quad \frac{\sigma_{max}}{1 + \lambda u^\top S} \qquad\qquad\qquad u_{max} S_{max} \text{ is positive}$$

For the right end point, we have

$$l := \max_k \frac{\sigma_k}{1 + \lambda u^\top S - \lambda u_k S_k}$$
$$< \quad \max_k \sigma_k \qquad\qquad\qquad 1 + \lambda u^\top S - \lambda u_k S_k > 1$$
$$= \quad \sigma_{max}$$

Therefore, $l$ is a reachable point. Moreover, any $\frac{\sigma_{max}}{1 + \lambda u^\top S} < x < l$ will still induce positive $y_k$ for any $k$ and we will also use this to prove that $x$ should not be smaller than $l$. Let

$$c := \arg\max_k \frac{\sigma_k}{1 + \lambda u^\top S - \lambda u_k S_k}$$

then we can substitute $l$ into the $x$ of $y_c$. It can be easily verified that $y_c = 1$ at such $l$. Since $y_c$ is monotonically decreasing in terms of $x$, any $x$ smaller than $l$ will cause $y_c$ to be greater than 1. At the same time, other $y_k$ is still positive as mentioned just above, so the $\sum_k y_k$ will be greater than 1. Thus, $l$ is a reachable left end point. $\qquad \square$

***Property* 2.** For any category $k$ such that $u_k > 0$, the set of strictly positive variables $y_k$ or $S_k$ can only grow during iterations of our EM algorithm for the loss (d) based on the collision cross-entropy.

*Proof.* As obvious from the E-step (14), it is sufficient to prove this for variables $y_k$. If $y_k = 0$, then the E-step (14) gives $S_k = 0$. According to the M-step for the case of collision cross-entropy, variable $y_k$ may become (strictly) positive at the next iteration if $\sigma_k = \sigma_{max}$. Once $y_k$ becomes positive, the following E-step (14) produces $S_k > 0$. Then, the fairness term effectively enforces the log-barrier from the corresponding simplex boundary making M-step solution $y_k = 0$ prohibitively expensive. Thus, $y_k$ will remain strictly positive at all later iterations. $\qquad \square$

# B Complete Solutions for M step

$$-\ln \sum_k \sigma_k y_k \;\; - \;\; \lambda \sum_k u_k S_k \ln y_k. \tag{d}$$

The main case when $u_k S_k > 0$ for all $k$ is presented in the main paper. Here we derive the case when there exist some $k$ such that $u_k S_k = 0$. Assume a non-empty subset of categories/classes

$$K_o := \{k \,|\, u_k S_k = 0\} \quad \neq \quad \emptyset$$

and its non-empty complement

$$\bar{K}_o := \{k \,|\, u_k S_k > 0\} \quad \neq \quad \emptyset.$$

In this case the second term (fairness) in our loss (d) does not depend on variables $y_k$ for $k \in K_o$. Also, note that the first term ( collision cross-entropy) in (d) depends on these variables only via their linear combination $\sum_{k \in K_o} \sigma_k y_k$. It is easy to see that for any given confidences $y_k$ for $k \in \bar{K}_o$ it is optimal to put all the remaining confidence $1 - \sum_{k \in \bar{K}_o} y_k$ into one class $c \in K_o$ corresponding to the larges prediction among the classes in $K_o$

$$c := \arg\max_{k \in K_o} \sigma_k$$

so that

$$y_c = 1 - \sum_{k \in \bar{K}_o} y_k \qquad \text{and} \qquad y_k = 0, \;\; \forall k \in K_o \setminus c.$$

506     Then, our loss function (d) can be written as

$$-\ln \sum_{k \in \bar{K}_o \cup \{c\}} \sigma_k y_k \;\; - \;\; \lambda \sum_{k \in \bar{K}_o} u_k S_k \ln y_k \tag{e}$$

507     that gives the Lagrangian function incorporating the probability simplex constraint

$$-\ln \sum_{k \in \bar{K}_o \cup \{c\}} \sigma_k y_k \;\; - \;\; \lambda \sum_{k \in \bar{K}_o} u_k S_k \ln y_k \;\; + \;\; \gamma \left( \sum_{k \in \bar{K}_o \cup \{c\}} y_k - 1 \right).$$

508     The stationary point for this Lagrangian function should satisfy equations

$$-\frac{\sigma_k}{\sigma^\top y} - \lambda u_k S_k \frac{1}{y_k} + \gamma = 0, \;\; \forall k \in \bar{K}_o \qquad \text{and} \qquad -\frac{\sigma_c}{\sigma^\top y} + \gamma = 0$$

509     which could be easily written as a linear system w.r.t variables $y_k$ for $k \in \bar{K}_o \cup \{c\}$.

510     We derive a closed-form solution for the stationary point as follows. Substituting $\gamma$ from the right
511     equation into the left equation, we get

$$\frac{\sigma_c - \sigma_k}{\sigma^\top y} y_k = \lambda u_k S_k, \qquad \forall k \in \bar{K}_o . \tag{f}$$

512     Summing over $k \in \bar{K}_o$ we further obtain

$$\frac{\sigma_c(1-y_c) - \sum_{k \in \bar{K}_o} \sigma_k y_k}{\sigma^\top y} = \lambda u^\top S \qquad \Rightarrow \qquad \frac{\sigma_c - \sigma^\top y}{\sigma^\top y} = \lambda u^\top S$$

giving a closed-form solution for $\sigma^\top y$

$$\sigma^\top y = \frac{\sigma_c}{1 + \lambda u^\top S}.$$

Substituting this back into (f) we get closed-form solutions for $y_k$

$$y_k = \frac{\lambda u_k S_k}{(1 + \lambda u^\top S)(1 - \frac{\sigma_k}{\sigma_c})}, \qquad \forall k \in \bar{K}_o .$$

Note that positivity and boundedness of $y_k$ requires $\sigma_c > \sigma_k$ for all $k \in \bar{K}_o$. In particular, this means $\sigma_c = \sigma_{max}$, but it also requires that all $\sigma_k$ for $k \in \bar{K}_o$ are strictly smaller than $\sigma_{max}$. We can also write the corresponding closed-form solution for $y_c$

$$y_c = 1 - \sum_{k \in \bar{K}_o} y_k = 1 - \frac{\sigma_c}{1 + \lambda u^\top S} \sum_{k \in \bar{K}_o} \frac{\lambda u_k S_k}{\sigma_c - \sigma_k}.$$

513 Note that this solution should be positive $y_c > 0$ as well.

514 In case any of the mentioned constraints ($\sigma_c > \sigma_k, \forall k \in \bar{K}_o$ and $y_c > 0$) is not satisfied, the
515 *complimentary slackness* (KKT) can be used to formally prove that the optimal solution is $y_c = 0$.
516 That is, $y_k = 0$ for all $k \in K_o$. This reduces the optimization problem to the earlier case focusing
517 on resolving $y_k$ for $k \in \bar{K}_o$. This case is guaranteed to find a unique solution in the interior of the
518 simplex $\Delta_{\bar{K}_o}$. Indeed, since inequality $u_k S_k > 0$ holds for all $k \in \bar{K}_o$, the strong fairness enforces a
519 log-barrier for all the boundaries of this simplex.

## 520 C   Optimization algorithms

---
**Algorithm 1:** Newton's method for M-step

---
**Input**   : $\{\sigma_k\}, \{S_k\}, \lambda, \epsilon$
**Output :** $\{y_k\}$
Initialize $x \leftarrow \max_k \frac{\sigma_k}{1+\lambda u^\top S - \lambda u_k S_k}$
calculate $f(x) \leftarrow \sum_k \frac{\lambda u_k S_k}{\lambda u^\top S + 1 - \frac{\sigma_k}{x}} - 1$
**while** $f(x) \geq \epsilon$ **do**
 | calculate $f'(x) \leftarrow \sum_k \frac{-\lambda u_k S_k \sigma_k}{(\lambda u^\top S x + x - \sigma_k)^2}$
 | $x \leftarrow x - \frac{f(x)}{f'(x)}$
 | calculate $f(x) \leftarrow \sum_k \frac{\lambda u_k S_k}{\lambda u^\top S + 1 - \frac{\sigma_k}{x}} - 1$
**end**
$y_k \leftarrow \frac{\lambda u_k S_k}{\lambda u^\top S + 1 - \frac{\sigma_k}{x}}$

---

---
**Algorithm 2:** Optimization for (11)

---
**Input**   : network parameters and dataset
**Output :** network parameters
**for** *each epoch* **do**
 | **for** *each iteration* **do**
  | Initialize $y$ by the network output at current stage as a warm start;
  | **while** *not convergent* **do**
   | E step: $S_i^k = \frac{y_i^k}{\sum_j y_j^k}$;
   | M step: find $y_i^k$ using Newton's method;
  | **end**
  | Update network using loss $\overline{H_2(y, \sigma)}$ via stochastic gradient descent
 | **end**
**end**

---

## 521 D   Self-supervision Loss Comparison

$$L_{CCE} \quad := \quad \overline{H_2(y, \sigma)} + \lambda \, KL(\bar{y} \| u) \tag{a}$$

$$L_{CCE+} \quad := \quad \overline{H_2(y, \sigma)} + \lambda \, KL(u \| \bar{y}) \tag{b}$$

|     | STL10        | CIFAR10      | CIFAR100-20   | MNIST       |
|-----|--------------|--------------|---------------|-------------|
| (a) | 92.32%(6.3)  | 73.51%(6.4)  | 43.73%(1.1)   | 58.4%(3.2)  |
| (b) | 92.33%(6.4)  | 73.51%(6.3)  | 43.72%(1.1)   | 58.4%(3.2)  |

Table 6: Using fixed features extracted from Resnet-50.

|     | STL10        | CIFAR10      | CIFAR100-20   | MNIST       |
|-----|--------------|--------------|---------------|-------------|
| (a) | 25.98%(1.0)  | 24.26%(0.8)  | 15.13%(0.6)   | 95.10%(4.2) |
| (b) | 25.98%(1.1)  | 24.26%(0.8)  | 15.14%(0.5)   | 95.11%(4.3) |

Table 7: With simultaneous feature training from the scratch. The network architecture is VGG-4.

# E  Experiments

## E.1  Network Architecture

The network structure of VGG4 is adapted from [17]. We used standard ResNet-18 from the PyTorch library as the backbone architecture for Figure 2. As for the ResNet-18 used for Table 4, we used the code from this repository [1].

| Grey(28x28x1)          | RGB(32x32x3)           | RGB(96x96x3)            |
|------------------------|------------------------|-------------------------|
| 1xConv(5x5,s=1,p=2)@64 | 1xConv(5x5,s=1,p=2)@32 | 1xConv(5x5,s=2,p=2)@128 |
| 1xMaxPool(2x2,s=2)     | 1xMaxPool(2x2,s=2)     | 1xMaxPool(2x2,s=2)      |
| 1xConv(5x5,s=1,p=2)@128| 1xConv(5x5,s=1,p=2)@64 | 1xConv(5x5,s=2,p=2)@256 |
| 1xMaxPool(2x2,s=2)     | 1xMaxPool(2x2,s=2)     | 1xMaxPool(2x2,s=2)      |
| 1xConv(5x5,s=1,p=2)@256| 1xConv(5x5,s=1,p=2)@128| 1xConv(5x5,s=2,p=2)@512 |
| 1xMaxPool(2x2,s=2)     | 1xMaxPool(2x2,s=2)     | 1xMaxPool(2x2,s=2)      |
| 1xConv(5x5,s=1,p=2)@512| 1xConv(5x5,s=1,p=2)@256| 1xConv(5x5,s=2,p=2)@1024|
| 1xLinear(512x3x3,K)    | 1xLinear(256x4x4,K)    | 1xLinear(1024x1x1,K)    |

Table 8: Network architecture summary. s: stride; p: padding; K: number of clusters. The first column is used on MNIST [25]; the second one is used on CIFAR10/100 [43]; the third one is used on STL10 [8]. Batch normalization is also applied after each Conv layer. ReLu is adopted for non-linear activation function.

## E.2  Experimental Settings

Here we present the missing details of experimental settings for Table 2 - 5. As for Table 2, the weight of the linear classifier is initialized by using Kaiming initialization [13] and the bias is all set to zero at the beginning. We use the $l_2$-norm weight decay and set the coefficient of this term to 0.001, 0.02, 0.009, and 0.02 for MNIST, CIFAR10, CIFAR100 and STL10 respectively. The optimizer is stochastic gradient descent with a learning rate set to 0.1. The batch size is set to 250. The number of epochs is 10. We set $\lambda$ in our loss to 100 and separately tuned the hyperparameters for other methods.

For Table 3, we use Adam [21] with learning rate $1e^{-4}$ for optimizing the network parameters. We set batch size to 250 for CIFAR10, CIFAR100 and MNIST and we use 160 for STL10. We report the mean accuracy and Std from 6 runs with random initializations. We use 50 epochs for each run and all methods reach convergence within 50 epochs. The weight decay coefficient is set to 0.01.

As for the training of ResNet-18 in Table 4, we still use the Adam optimizer, and the learning rate is set to $5e^{-2}$ for the linear classifier and $1e^{-5}$ for the backbone. The weight decay coefficient is set to $1e^{-4}$. The batch size is 200 and the number of total epochs is 50. The $\lambda$ is still set to 100. We only use one augmentation per image, and the coefficient for the augmentation term is set to 0.5, 0.2, and 0.4 respectively for STL10, CIFAR10, and CIFAR100 (20).

As for the semi-supervised settings, we made two changes compared to the above. First, we added the cross-entropy loss on the labeled images and set the weight to 2, and separately tuned the hyperparameters for other methods. Second, the pseudo-labels on the labeled images are constrained to be the ground truth during the optimization.

---

[1] https://github.com/wvangansbeke/Unsupervised-Classification

