# OpenReview forum: "Collision Cross-entropy for Soft Class Labels and Entropy-based Clustering"
_NeurIPS.cc/2024/Conference — Submitted to NeurIPS 2024_

### Official Review · Reviewer_Puk1 · 2024-07-08

**Soundness:** 3
**Presentation:** 3
**Contribution:** 1
**Rating:** 4
**Confidence:** 3

**Summary:**

Soft labels are often used to represent ambiguous/noisy/uncertain targets in classification, particularly in self-labelled clustering, where pseudo-labels are estimated together with model parameters.
The authors propose an alternative to Shannon cross-entropy for a loss term, called the collision probability.
This term arises as a limiting case of a Renyi entropy, or as a probability that two random variables are equal.
The collision cross entropy admits several advantageous properties: it is robust to large deviations in the target data, it agrees with Shannon cross-entropy for one-hot labels, it is symmetric, and points that are labelled as uniform distribution have no contribution to training.
The authors provide an EM algorithm for pseudo-label estimation and show state of the art results.

**Strengths:**

- The paper is **very well written**, providing strong intuition and flowing prose. The intuition in Figure 1 is helpful, especially in showing that the proposed measure is robust to large target errors.
- The main technical element appears to be an EM algorithm for solving the clustering problem obtained by using a collision cross-entropy in place of the Shannon entropy (which a swap in the arguments between equations (10) and (11)). This algorithm appears to be **technically sound**, and guarantees convergence of the subproblem in the M step.
- The proposed term has **several nice properties** (as I mentioned earlier). Tt is robust to large deviations in the target data, it agrees with Shannon cross-entropy for one-hot labels, it is symmetric, and points that are labelled as uniform distribution have no contribution to training.

**Weaknesses:**

- My main concern is that **conceptually, the contributions are rather limited**. Generalisations of entropy are well-known, and as far as I understand (correct me if I am wrong), the main contribution is that authors use a different measure of entropy to Shannon entropy inside existing formulations. This leads the authors to investigate EM-algorithm and empirical performances, but as the paper is currently written (see further comments below), I cannot see whether these EM-algorithm and empirical performance benefits are actually real and beneficial. I also do not understand why this particular notion of entropy was used, compared with the other spectra of entropies.
- An incomplete review of relevant generalized formulations of entropy is provided. This is not a weakness per se, however **perhaps the title in section 2.2 could be changed to something like Renyi Entropy**. Similarly tone down the discussion of generalised entropy measures throughout the paper. Alternatively, the authors might consider expanding their discussion and including more well-known entropy measures. For example, see section 8 and 11 of [1].
- The bold numbers in table 2 require clarification. The caption doesn't mention the number of trials (however the text mentions 6 trials). Compared with MIGD, excluding MNIST, due to the high variance in the trials, the results do not appear to be **statistically significant**. Perhaps the authors could consider running more trials and performing a significance test, and/or also bold relevant entries in MIGD.
- As above for Table 3, 4 and 5.
- It is **not clear how long the method takes to run** compared with competitors. Does the EM algorithm outperform a naive marginalisation of the log likelihood (using e.g. MC), both in terms of time and in terms of predictive performance?
- Related to the above, is the reason for specialising on $\alpha \to 1$ because it allows for the EM algorithm? If you consider other values of $\alpha$, how do the results compare in terms of time and performance. Or is this setting intractable?


[1] Generalized Thermostatistics, Jan Naudts, 2011.


Minor:
- The text in the tables is too small to read without zooming in a lot.
- Recommend less active tense in the abstract: "In case of soft labels y, Shannon’s CE teaches the model predictions σ to reproduce the uncertainty in each training example" could be "In case of soft labels y, Shannon’s CE results in model predictions σ which reproduce the uncertainty in each training example".

**Questions:**

Please see weaknesses above.

**Limitations:**

The author checklist appears to be incomplete. The authors answer NA to "Does the paper discuss the limitations of the work performed by the authors?", without a justification. I do see a small discussion around local minima and numerical instability towards the end of section 4, but I think these could be further elaborated on.

---

> ### Author Rebuttal · Authors · 2024-08-07
>
> **Weakness 1: conceptually, the contributions are rather limited**\
> To the best of our knowledge, collision cross entropy (9) is a new concept if the focus is on "collision", though collision entropy (6) is standard. Note that Renyi's work generalizes entropies and divergences, but not cross-entropy. There are several different later extensions of cross-entropies, some of which agree with (9) as discussed in lines 198-206. However, our motivation for (9) as a probability of "collision" is novel, see lines 184-192. We did not see this in prior work. Neither we known any prior work using CCE (9) as a loss for training with noisy labels, which is the key contribution of our work.
>
> The reviewer's summary of our contributions makes them sound too obvious. However, it matters that it was not done before. Many ideas become obvious after they are presented and explained. However, it was not easy to discover them and to make them obvious. Based on some other reviews, we have to work more to make them so obvious.
>
> In any case, are there references to prior work where CCE is used for network training with noisy labels? Maybe we missed something.
>
> **Weakness 2: emphasizing Renyi entropy**\
> We are big fans of the work of Renyi and provide sufficient information and credits in section 3 (e.g. see lines 162 -168, 198-206). However, there is a reason we did not put the name Renyi in the title of section 2.2. The ultimate goal of this section is to get to cross entropies, like (8), The issue is that Renyi did not generalize cross entropy, only entropy and divergence. Our limited knowledge of the matter does not give a clear answer for why it is so. Moreover, we found that later literature introduces multiple inconsistent generalizations of cross-entropy, which we do not want to call Renyi cross-entropy to avoid confusion. This also explains why we are careful when discussing such extensions concentrating them in a brief isolated paragraph 198-206.
> Unlike Renyi's work, we did not see a simple clear motivation distinguishing any of these generalized cross-entropies. Moreover, we did not find such generalized cross-entropies too useful, except for our very simple CCE variant with our clear interpretation via collision (see more in the last point on "choosing a" at the botting of this rebuttal).
>
> **Weakness 2: Tables 2,3,4,5, statistical significance, etc**\
> As common for similar tables in the literature, the boldface font in Table 2 indicates the best result (in each column), We will clarify this. About adding more experiments and more recent architectures... we use the same benchmarks/architectures as in related prior works. Also, we reported results (including stdiv) consistently with the related prior work.
>
> While we can extend experiments for our method, it is not easy to find comparable numbers for other works, Moreover, our main goal was not to break SOTA, though we do well. Our goal was to demonstrate the properties and benefits of CCE, which we believe is done sufficiently.
>
> **Run time vs competition**\
> We took efficiency seriously, which is the main motivation for our EM solver since it addresses a convex optimization subproblem for our self-labeling loss (11). This is illustrated by the comparison with a naive solver in Table 1. Based on our efficient EM solver, our running time is on par with the solver proposed in [16] for (11). Running time is not a problem for our method.
>
> **choosing a**\
> We agree that it is interesting (for completeness) to explore the order parameter "a" for cross-entropy extensions (we found several inconsistent variants). This is similar to exploring Lp norms in the context of least squares. However, our work is directly motivated by the maximization of the probability of collision between random variables representing the unknown label and predicted class. We do not see a similar clear probabilistic motivation for other variants of cross entropy out there.
>
> It is also true that other forms of cross entropy are based on more complex formulas that do not easily lead to an efficient solver. We looked at this at some point. We even played with a naive solver to see if there was any empirical gain. We had only anecdotal preliminary numbers, but they did not look promising. Nothing to write home about :( Maybe we missed something, but we decided to leave it for future work and/or for other people.
>
> We are very happy with collision cross entropy due to its simplicity and clarity of probabilistic interpretation and motivation, desirable numerical properties (Fig 1), and excellent robustness to noise (Fig 2). We are excited to share these ideas with the community.

---

> > ### Comment · Reviewer_Puk1 · 2024-08-12
> >
> > Thanks for responding to my review.
> >
> > - Limited contributions. I understand that the authors have introduced a new loss function (collision cross-entropy), and studied the EM algorithm in this context, developed various algorithms, and benchmarked their results. I am not aware of any prior works that do this. However, there are uncountably infinitely many notions of entropy, and I do not think each of them warrant a paper that can be labelled as conceptually novel. What then, does an interesting paper in this space look like? Perhaps one can start with a suitably general notion of entropy (as per my hint with the provided reference), and then study properties of all of these notions of entropy in the context you are looking at. Some properties might be whether or not certain notions are tractable, whether they lead to other probabilistic interpretations (e.g. collision), whether they have been studied before, whether they allow for strong performance, and what are there relative strengths and weaknesses? At the moment, I am seeing a good start to this, by focusing on one entropy, but I am unconvinced from the evidence presented in the paper that this single choice is a very interesting choice. Convince me and other readers, by discussing and trying other entropies.
> > - Renyi Entropy. My suggestion was that your label of ``generalised entropy'' is inappropriate, given you study only one form of entropy (whether cross-entropy was studied by Renyi or not is a separate matter). Either change the label to something more specific than generalised entropy, or expand your literature to include other forms of generalised entropy.
> > - Bold tables. It may be common for similar tables to be bold to indicate the best result, however this is bad practice if the results are not meaningfully better. One way to measure meaningfulness is to take multiple runs, and observe significantly more variation between the methods than between the runs. The current presentation does not allow for this. I am fine that your goal was not to break SOTA. Just report the result that you obtained, and if the difference is not meaningful, no need to claim that they are.
> > - Run time. Thanks very much for pointing me to table 1, which I missed. Excellent comparison, clearly showing your result is faster. I might suggest making the font bigger in a future version of the paper.

---

> ### Author Response · Authors · 2024-08-12
> **Response to the first bullet**
>
> We thank the reviewer for continuing the engaging discussion, which we hope can clarify things. We are also glad that the reviewer agrees that no prior work does what we claim our main contributions are, that is, the introduction of a new form of cross entropy as a loss function for network training with noisy labels and an efficient algorithm for this loss. It seems that the reviewer doubts the significance of our contributions (the first bullet above). We would like to convince the reviewer about the significance. We also separately respond to the points in bullets two and three, though they seem less consequential.
>
> > Limited contributions... I am unconvinced from the evidence presented in the paper that this single choice (collision cross entropy) is a very interesting choice. Convince me and other readers, by discussing and trying other entropies.
>
> Summarized in our own words, the first bullet says that our cross-entropy choice is only one of an infinite (uncountable) number of different generalized cross-entropies, **all** of which should be studied in our context (network training with noisy labels). In our minds we have no doubts that the introduced collision cross-entropy (CCE) is very interesting, which is why we are eager to share it with the community.  Here is a summary of how we see the potential significance of our work for the community:
>
> A. First of all, the very fact that our work proposes to look beyond Shannon’s entropy in the context of network training (unsupervised or with uncertain/noisy labels) could be important. For example, to the best of our knowledge, all prior work on deep clustering and self-labeling is “stuck” with Shannon’s entropy or cross-entropy.  We explain **numerical** (Fig.1), **empirical** (Fig.2), and **conceptual** limitations of Shannon’s entropy (the difference between equality of distributions and random variables discussed below (9) that we plan to emphasize). We propose CCE as a well-motivated mathematically solid alternative that can address such limitations. **While CCE is only one of the possible alternatives (e.g. mentioned in l.198), our detailed study of this single example is enough to demonstrate the existence (of the whole class) of alternatives to Shannon that can improve network training**. We believe that this is new knowledge for the community and hope that more researchers will be inspired to study other examples of generalized entropies.
>
> B. We point out a unique property motivating CCE - **maximization of (the log of) probability of equality between (random variables) predicted class and unknown true class**, see below (9). This replaces the property of Shannon CE - the enforcement of equality of the distributions for these random variables. The difference is significant and consequential. The proof is based on very simple algebra (dot product is the sum of probabilities of equal outcomes), but we see no technical reason that it may work for other (more complex) general cross-entropy formulae. This property **makes CCE stand out**. This CCE property also conceptually explains (see below (9)) why CCE resolves numerical and empirical limitations of Shannon’s cross-entropy illustrated in Fig.1 and Fig.2. These are not by chance.
>
> C. In particular, the conceptual new properties of CCE lead to strong empirical improvements over standard Shannon's cross-entropy convincingly demonstrated by the "clean" test in Fig.2. So far the reviewer has not acknowledged this test. We think the **example in Fig.2 is very important for appreciating the significance of the new general ideas and CCE in particular**. Do you have any questions or comments about this example?
>
> D. Our preliminary evaluation of some other general forms of cross-entropy in [30,32,46] could not improve the results for CCE (e.g. those in Fig 2). These results are far from conclusive, which is why they are not in the paper. However, this worked as (anecdotal) evidence for the potential special status of CCE, already supported by the (unique) conceptual property discussed above.
>
> Finally, we agree with the reviewer that a paper focused on a random special case of a well-known general class of extensions may not be interesting. But, the points above argue that neither the general class of (Renyi) CE is well-known in network training, nor CCE is a random choice. Also, we find unrealistic the reviewer’s expectation that a full exhaustive study of conceptual, algorithmic, and empirical properties of all (uncountable) number of generalized entropies should be done in the scope of one NeurIPS paper. There is enough room for future research and we hope the community will be interested in helping. We are excited that other reviewers already suggested some new applications for the general ideas proposed in our work.

---

> ### Author Response · Authors · 2024-08-12
> **Response to the second and the third bullets**
>
> > Your label of “generalised entropy” is inappropriate. …suitably general notion of entropy, as per my hint with the provided reference (to the book on “Generalized Thermostatistics” by  Jan Naudts)
>
> We downloaded the book. As promised by the reviewer, Chapters 8 and 11 discuss general entropies and divergences (relative entropies) in the general context of **Thermodynamics**. We believe that generalization of entropies done by Renyi in the context of **Information Theory** (reviewed in our Sec 2.2) is significantly more directly relevant for our work, as we hope you can agree.  Also, the motivations in these two areas are sufficiently different. Indirectly confirming this, the book has no references to the famous works of Renyi or his followers. The two areas can be related, e.g. Shannon was inspired by entropy ideas in Thermodynamics. But discussing this relation did not even make it into the book; it is an even bigger stretch to expect this from our conference paper.
>
> In short, we disagree with the suggestion in your original review to expand our discussion of generalized information-theoretic Renyi entropies in Sec 2.2 by generalized entropies in thermodynamics. However, we will happily cite this book in the beginning of Section 2.2 to indicate the existence of other generalizations outside the context of information theory. Moreover, we can also change the title of Section 2.2 to “Generalized Entropy Measures in Information Theory” to explicitly state its limited scope. We hope you will find this simple fix to be acceptable.
>
> > Bold font in the tables…
>
> We did not know that bold font numbers in the Tables could be interpreted as a claim of statistical significance. We just followed a practice that we thought was standard “these days”. We do not mind removing the bold font. Alternatively, we can explicitly state that there is no claim of statistical significance and that we just follow a common practice for experimental design and reporting. Perhaps we can ask for the recommendation of the area chair on this matter once the final decision is made about the paper. We hope your decision about the paper is not affected by this font issue.

---

> > ### Comment · Reviewer_Puk1 · 2024-08-12
> >
> > I appreciate your response, and see the value in your work (I raise my score by 1), but think this paper needs more work to give it the attention it deserves. *If* you would like to action my feedback, you will need some more time than the review cycle allows to properly digest this literature, if it was not already known to you beforehand. But of course there would be other ways to improve the paper, if you don't want to consider my feedback.
> >
> > There are many works in that look beyond Shannon's (cross)-entropy in the context of clustering, and machine learning more generally. I am not suggesting you need to enumerate uncountably many infinite entropies, just discuss them, study some of them, and place your entropy of interest in the broader context of established methods. (note that the classical cross-entropy is related up to a constant to the "forward/backward" $f$-divergence induced by $f(z) = z\log z$ or $f(z)=-\log(z)$, depending on which argument is held constant, and references to $f$-divergences allow for the generalisations I am suggesting).
> >
> > - Clustering above Exponential Families with Tempered Exponential Measures, AISTATS 2023
> > - Geometry of q-Exponential Family of Probability Distributions
> > - Amari's book,
> > - or works by Nielsen
> >
> > The link between physicist's entropy and information theorist's entropy is well-known, and while the reference I provided is in the context of thermodynamics, many works have investigated it in the context of information geometry (see the works that cite it in Google Scholar).

---

> ### Author Response · Authors · 2024-08-13
>
> We thank you for all your feedback. We will look more carefully into the provided references and will try to action your feedback.
>
> We have some more thoughts based on some new references and would like to bounce them back (though we might be off since we only had a very quick look). In general, our impression is that your comments mainly concern limitations in our description of related work. We certainly care about related work and will do our best to add references. However, we feel that the provided references are only indirectly related to the main topic of our work - network training with noisy labels in the context of self-labeled "deep clustering". The latest references provided by the reviewer helped us to better see a common theme - the use of entropy/divergence measures for clustering. However, it also became more evident that we might be discussing two significantly different groups of entropy clustering algorithms. Somewhat informally, we can refer to them as discriminative and generative.
>
> **A. [discriminative entropy clustering]** our work is in the group started by MacKay et al [3] at NIPS 1991.
> We refer to it as "discriminative" since it explores unsupervised loss functions for discriminative softmax models (networks).
> MacKay starts from a Mutual Information criterion and derives entropy-based decisiveness and fairness losses applicable to soft-max models. This is why **our work concerns network training**. This also explains why our work cares about **information-theoretic entropy** (e.g. Renyi entropies) that is directly suitable for evaluating multi-class decisions (categorical distributions).
>
> **B [generative entropy clustering]** This is a much older group related to K-means. Works like AISTATS 2023 (we only had a quick look, so we could be wrong) discuss generalized K-means where entropies/divergences can be used as general measures of cluster compactness (as an alternative to the sum of squared errors corresponding to the entropy under Gaussinity assumption). We call this "generative clustering" since the entropy evaluates the properties of the data (density) in each cluster, rather than the property of the prediction model (posterior) outputs. This also explains why **the references provided by the reviewer seem unrelated to networks** and focus on **entropy from statistical physics** that is directly suitable for evaluating chaos in the data.
>
> While the relation or differences between groups A and B are interesting (we are working on a PAMI submission discussing this in more detail), we do not think that generalized entropies discussed in group [B] are reducing the significance of our work motivating group [A] to use entropies other than Shannon's.
>
> Does this make sense to the reviewer?
>
> If it does, we can discuss this relation in Sec 2.1 and place your references there.
>
> If it does not, please do not lower your rating. We will think harder about your references :) Or maybe you can give us another hint on how to relate your references to our work.
>
> Thanks again.

---

### Official Review · Reviewer_C7Wi · 2024-07-14

**Soundness:** 2
**Presentation:** 3
**Contribution:** 2
**Rating:** 4
**Confidence:** 4

**Summary:**

The paper introduces the concept of collision cross-entropy (CCE) as an alternative to Shannon's cross-entropy (SCE) for self-labeling in the context of unsupervised and semi-supervised learning. The primary motivation is to address the limitations of SCE, especially its sensitivity to label noise and uncertainty. CCE aims to enhance robustness to such uncertainties by defining a probabilistic interpretation that encourages collision events between predicted and true distributions. The paper provides theoretical foundations, describes an EM algorithm for efficient optimization, and presents experimental results demonstrating the superior performance of CCE over SCE on the task of deep clustering.

**Strengths:**

Originality
-The paper introduces a novel loss function, the collision cross-entropy, which is well-motivated by the need to handle soft and uncertain labels in classification tasks, particularly in self-labeled clustering. The idea of maximizing the collision probability is distinct from the traditional approach of minimizing the (implicit) KL divergence between distributions.

Quality
-The paper provides a solid theoretical foundation for the collision cross-entropy, including its properties and relationship to other entropy measures. The derivation of an efficient EM algorithm for pseudo-label estimation further strengthens the paper's technical contribution.

Clarity
- The paper is generally well-written and organized. The motivation, theoretical analysis, and experimental results are presented clearly. The authors provide sufficient details for an expert reader to understand and potentially reproduce the work.

Significance
- The proposed collision cross-entropy has the potential to be a valuable tool for handling soft and uncertain labels in various machine learning tasks.

**Weaknesses:**

Quality
- The superiority of CCE seems to hinge on making the model capture the same "decisions" as the target distribution, without forcing the model to capture the entirety of the distribution, as well as de-weighting target distributions which are not spiky. While the properties of the loss are clear, it is not self-evident to me that the properties *of the loss function* translate into necessarily *better properties for models*, both as a function for training a classification model directly or for clustering.
- In addition, the experiments were conducted on fairly old architectures (VGG, ResNet) and small datasets. Often improvements on small datasets do not translate into improvements on larger-scale models. I would encourage the authors to examine for full imagenet dataset at the very least. This also open up the capability to look at various robustness / calibration properties of the models on the various corrupted forms of ImageNet.

Clarity
- Certain sections, the task to which this method is applied and the desired model properties for the task could be more clearly explained. It took me a while to get my head around the deep clustering task which the authors are solving.

Significance
- The impact of CCE on real-world applications beyond the presented datasets and tasks could be further elaborated. This notion that CCE is better for noisy pseudo labels immediately suggests to me examining it as a loss function for doing distillation / noisy teacher-student training of a model on a pseudo-labelled corpus of data, however, I didn't see any links to the area of distillation / teacher-student training within this paper.
- The significance would be bolstered by demonstrating CCE's performance on larger scale, more diverse and challenging datasets.

**Questions:**

- **Impact of Soft Label Quality:** How does the quality of the soft labels (e.g., the degree of uncertainty) affect the performance of the collision cross-entropy compared to the standard cross-entropy? Are there scenarios where the standard cross-entropy might be preferable? How strongly does this problem scale with the number of classes?
- **Numerical Stability:** Are there any numerical stability issues in optimizing CCE? I can imaging the taking the log of the sum of a product of small numbers can cause problems.
- **Applicability to Other Tasks:** The paper focuses on self-labeled clustering. Can the collision cross-entropy be applied to other tasks involving soft labels, such as semi-supervised learning, distillation and teacher-student training? What modifications or adaptations might be necessary? How would this work on sequence-based pseudo-labels, such as are encountered in ASR, machine translation and language modeling. It seems to me that CCE could be tried as a drop-in-replacement loss function.

**Limitations:**

**Strengths:**
- The paper acknowledges the need for robustness to label noise and addresses this effectively through CCE.
- The paper briefly mentions the potential increase in privacy disclosure risk with larger synthetic datasets but does not elaborate on this limitation or discuss potential mitigation strategies. It would be beneficial to include a more detailed discussion of the privacy implications of the proposed method and any potential negative societal impacts.

**Weaknesses:**
- The discussion on limitations could be more explicit, particularly regarding any assumptions made and potential edge cases where CCE may not perform optimally.
- The experimental evaluation uses very old model architectures (VGG, ResNet) and small datasets (CIFAR-10, CIFAR-100, MNIST, STL-10) which feature images only as large as 96x96 pixels. I would be curious whether the advantages of this method translate to high-dimensional image data with more classes, and on more modern, transformer-based  architectures (eg: ViT) .
- Similarly, could the author's consider extending this approach to deal with pseudo-labelled sequence data, for language models or for translation models, for example?

---

> ### Author Rebuttal · Authors · 2024-08-07
>
> **Weakness 1: it is not self-evident to me that the properties of the loss function translate into necessarily better properties for models, both as a function for training a classification model directly or for clustering.**\
> Probably the clearest evidence that CCE is a better loss for learning from noisy uncertain labels directly translating into better model training is demonstrated by the example in Figure 2, which we hope is fully explained in the captions. If it does not help as much as we hope, please clarify your concern a bit more. We will try to further elaborate at the next phase.
>
> **Weakness 2: experiments**\
> We use the same benchmarks and architectures that are used in most (if not all) of the related prior works. It is hard to start using more recent architectures as we will have nothing to compare to. Most prior works do not have easily available code.  Also, please note that (as explained to other reviewers above), while our general self-labeling framework based on CCE is doing very well w.r.t. SOTA, we think that our main contribution is the introductions of a new general concept - collision cross entropy - which can be used in many applications due to its generality and clear conceptual properties motivating its use.
>
> **Weakness 3: clarity of deep clustering**\
> We agree that the relatively recent term (or jargon) "deep clustering" could be better explained in the background section 2. This is easy to fix though.
>
> **Weakness 4: significance should be elaborated**\
> We hope that significance could be better understood with the example in Figure 2, which we now see belongs to page 2 in the summary of contributions, rather than on page 3 where it is lost in the overview of self-labeling. Please let us know what you think about Figure 2.
>
> Also, we agree that distillation could be another interesting application for collision cross entropy. We think it could be mentioned in FIgure 2 and in the list of potential other applications, e.g. at the end of the summary of contributions or in the conclusions and extensions section. Thanks for this great suggestion, as well as for the suggestions for the language models at the end of your review.
>
> **Weakness 5: more datasets**\
> Please have mercy :) The number of experiments already done for this paper is fairly significant and comparable to closely related work. We like this topic very much and will work on more applications and datasets, but it takes a lot of time. We can't do it in one shot. Please let us publish the main idea (collision cross entropy). There is already enough to talk about, as you probably agree based on the length of your review. We will eventually expand its applications. Perhaps others can help too.
>
> **Question 1: impact of soft label quality?**\
> See Figure 2.
>
> **Question 2: numerical stability issues for CCE?**\
> We saw nothing extra to what exists in standard cross entropy. We can mention this. Thanks
>
> **Question 3: applications to other tasks**\
> As already discussed above, we agree that this should be better highlighted perhaps at the end of the summary of contributions. Very good idea that is also easy to implement. We will also think about your other suggestions for "pseudo-labelled sequence data, for language models, or for translation models". Thanks.
>
> **Weakness 6: limitations?**\
> We did not see examples where CCE works worse than standard cross entropy. So, we are not sure what to write on the limitations.

---

> > ### Comment · Reviewer_C7Wi · 2024-08-13
> > **Response**
> >
> > I appreciate the author's response. However, I do feel quick strongly about using old models, especially if  (as pointed out by other reviewers), the models' don't reach the level of performance on established tasks which they reached in prior work.
> >
> > In light of this, as well as the extensive discussions with other reviewers, I will keep my score as is.

---

> > > ### Author Response · Authors · 2024-08-14
> > > **on newer methods**
> > >
> > > Below we copy our response to pteP on a similar issue of comparison to more recent results.
> > >
> > > >SOTA experiments in [DTC+2023] paper
> > >
> > > The main contribution of our paper is a new (for network training) version of cross entropy (CCE) as an alternative to standard Shannon's cross-entropy. Our validation of CCE wrt standard entropy was focused on comparison with other methods using entropies (including SCAN and others in our Table 4). These and other results (including our Fig 2) empirically validate the advantages of the proposed CCE. Do you see any specific reasons to disagree?
> > >
> > > Our paper is not focused on SOTA. But, it does not mean that we do not care. We thank the reviewer for pointing out some references, e.g. [DTC], that we can happily reference, discuss, and point out their better numbers. Please note that achieving SOTA in clustering requires many tricks. For example, there are three versions of SCAN (see their original paper) and the results get better and better as more tricks are added. Their pure clustering part using entropy losses achieves only 81.8 ACC on CIFAR10, but with other tricks it goes up to 87.4. The result in DTC paper (MLC) achieves 92.2 ACC using MoCo pertaining (thanks for clarifying this). Strangely, they do not show their MLC result for SimCLR pretraining in the first block in their Table 4. This result would have been the most relevant to compare with our number (83.3) since we use SimCLR pretraining. Also, note that SCAN-SimCLR number 87.6 reported in DTC paper is higher than the number we compare with for SCAN in our paper (81.8). We selected the SCAN result without any extra tricks for a fair comparison.
> > >
> > > In short, we do not believe that it is easy to make any solid conclusions about SOTA based on Table 4 in DTC paper, but we will gladly add this reference and discuss the challenges in establishing the truth in the state of the art.

---

### Official Review · Reviewer_VSk7 · 2024-07-15

**Soundness:** 3
**Presentation:** 2
**Contribution:** 2
**Rating:** 5
**Confidence:** 2

**Summary:**

The paper focuses on the choice of the loss function in problems with soft distributions of the labels, in particular in the context of pseudo-labeling for unsupervised or self-supervised problems such as clustering. In sections 1-2 the paper gives a thorough review of existing practices and relevant theoretical research. In section 3 the paper proposes a new collision cross-entropy loss as a replacement of the standard Shannon loss, and discusses various aspects of this new loss. In section 4 the paper proposes a new EM algorithm for pseudo-label estimation in connection with the new loss. Finally, in Section 5 the new algorithm is experimentally compared with existing ones and is shown to outperform them.

**Strengths:**

The paper is generally well-written. Sections 1-3 contain a thorough discussion of entropies and losses suitable for self-labeled clustering, with abundant references. The main point of the paper, the new collision cross-entropy loss, is well explained and motivated. The paper provides an experimental comparison of the proposed algorithm with alternatives and shows its significant advantage.

**Weaknesses:**

I'm confused by mixing the discussion of losses and EM algorithms in section 4. The bulk of the paper is focused exclusively on the advantages of the proposed new collision cross-entropy loss. The main claim in the abstract and introduction is that the proposed loss is better than the standard Shannon loss. The EM algorithm is mentioned only in the last line of abstract, as if in passing. However, the experimental comparison in section 5 obviously crucially depends on the EM algorithm proposed in section 4. How can we tell if the experimentally demonstrated advantage is due to the new loss or the EM algorithm? Since the main claim is about the superiority of the loss, why not just take any existing soft-labeled clustering algorithms and replace the standard Shannon loss by the proposed new loss? In my opinion, the lack of such a direct comparison substantially weakens the main claim of the paper. The advantage shown experimentally is good, but the conceptual takeaway may be misleading.

I found section 4 on the EM algorithm harder to read relative to the other sections (in fact, I'm not familiar with such algorithms and not even sure what EM stands for - apparently Expectation-Maximization, but this acronym is not explained in the paper). In constrast to the other sections, this one seems to assume familiarity of the reader with related algorithms. I didn't understand, for example, how equation (14) (E-step) was derived.

Another weakness I see is that the strongest results of the paper are largely experimental (not counting general arguments and auxiliary theoretical constructions in Section 4), but, as far as I understand, they are not easily verifiable since the code is not open-sourced.

**Questions:**

N/A

---

> ### Author Rebuttal · Authors · 2024-08-07
>
> **Weakness 1: is it the new loss or EM algorithm**
> Standard cross-entropy loss is used in [16] in the context of self-labeling (also using a specialized solver, only for efficiency due to convexity).  Standard cross-entropy is used without self-labeling in [3], which is evaluated in [16] and some earlier self-labeling methods.
> Our main baseline is [16] as the key change from their (5) to our self-labeling loss (11) is in the collision cross entopy.
> We also reversed the order in the fairness term, but it mainly allowed us to develop an efficient EM solver.
> The corresponding labeling subproblem in (11) or (5) is convex, and our EM for (11) is needed mainly for efficiency (see Table 1).
>
> **Weakness 2: what is EM?**
>
> We assumed that variational inference and EM algorithms are sufficiently standard for ML community. One standard example of an EM algorithm is typically covered when GMMs are taught. E.g. it is covered in Bishop's and most other standard textbooks for ML. Our Section 4 is self-contained, but it might be terse for a reader who has not seen EM algorithms before. BTW, this is cool stuff from optimization point of view and we strongly recommend looking into this. GMMs are also important to know in the context of clustering as this can be seen as a glorified soft K-means. We hope that the reviewer could accept our excuse about the brevity of Section 4 given the space limitations. We can help by answering specific questions in the next phase.
>
> **Weakness 3: what is the strongest contribution?**\
> we believe that the most significant contribution of our paper is the introduction of collision cross-entropy as a general concept (see Fig 2 illustrating its usefulness in a completely different context from clustering). We are also happy about the proposed self-labeling method for clustering using this collision cross entropy, but we mainly care about it as an interesting example.
>
> In any case, we will provide the code for our EM algorithm (it already exists), though it is also not hard to reimplement.

---

> > ### Comment · Reviewer_VSk7 · 2024-08-11
> >
> > I thank the authors for their reply. I'm keeping my score as is.

---

### Official Review · Reviewer_pteP · 2024-07-17

**Soundness:** 3
**Presentation:** 2
**Contribution:** 3
**Rating:** 4
**Confidence:** 3

**Summary:**

This paper studies the loss function for soft class labels and entropy-based clustering. In particular, it introduces a new loss function called 'collision cross-entropy' as an alternative to Shannon's cross-entropy when class labels are represented by soft categorical distributions. The motivation for this new loss function is to handle ambiguous targets/labels in classification. The authors provide an EM algorithm for pseudo-label estimation and conduct experiments to demonstrate that this approach leads to improvements in classification accuracy when models are trained with soft, uncertain targets.

**Strengths:**

- The proposed collision cross-entropy may have advantages over Shannon's cross-entropy when handling soft labels in certain scenarios.
- Through experiments, the authors demonstrate that the proposed method achieves better robustness to label uncertainty, which is important for self-labeled clustering methods.

**Weaknesses:**

- [**Theory-1**] The main contribution of this paper is proposing the new loss function 'collision cross-entropy'. However, there is not much theoretical analysis about this loss function. From the current paper presentation, the Eqn. (9) can be interpreted as a modified version (or inspired by) Eqn. (6). For example, by minimizing the new objective for learning linear models, could this new loss lead to the right linear classification model?

- [**Theory-2**] For the EM algorithm, is there any convergence analysis for the EM algorithm proposed in this paper?

- [**Experiments**] State-of-the-art for comparison. The methods for comparison in Table 1/2/3 are not very recent. It is possible that the previous methods still work well and be the state-of-the-art. However, I found some recent papers could achieve much better results, for example, the ACC on CIFAR10 of [DTC+2023] is 92%+, however the result in this paper is <84%.


[DTC+2023] Unsupervised Manifold Linearizing and Clustering. Tianjiao Ding, Shengbang Tong, Kwan Ho Ryan Chan, Xili Dai, Yi Ma, Benjamin D. Haeffele. ICCV 2023.

**Questions:**

- What's the definition of $u$ in Eqn. (4) (5) (10)? The presentation of this paper could be further improved, especially the part related to loss functions.

- The font size of Eqn. (16) is a bit odd.

- What's the first letter 'y' in Line 3? If it is $y$, then make it consistent with the label $y$ in Line 6.

**Limitations:**

See Weaknesses.

---

> ### Author Rebuttal · Authors · 2024-08-07
>
> **Theory-1**\
> We provide theoretical motivation for (9) on lines 184 - 192. It is a loss maximizing the probability of collision (equality) between two random variables: unknown true class (represented by solf-label distribution) and predicted class (represented by soft-max prediction/distribution). This is unlike the enforcement of equality between the two distributions represented by standard cross-entropy (8), see also Fig 1. The only difference between (6) and (9) is that (6) maximizes the probability of collision for two identically distributed random variables, while (9) allows different distributions. We agree that theoretical/probabilistic motivation is important. We will better emphasize this part, e.g. by creating a boldface paragraph or a subsection. Thanks for bringing this out.
>
> Also, in the context of fully labeled data where all labels are known (one-hot distributions), collision cross entropy (9) is equivalent to standard cross entropy. Thus, it has the same Bayesian consistency properties and guarantees correct classification (e.g. in the linear case).
>
> **Theory-2**\
> Our EM algorithm is derived following standard ideas in variational inference, exactly as EM for GMM. In particular, (13) is a tight upper bound for our loss (12). Two consecutive E and M steps are guaranteed to decrease the original loss (12), as in general bound optimization methodology (e.g. see Boyd). We will gladly clarify this.
>
> **Experiments**\
> While the results in [DTC+2023] are based on the same backbone (resNet18) as ours in Table 4, significant differences could be in initialization that are critical for SOTA comparisons in deep clustering. For example, we use pertaining from SCAN [45].
> We did not find the details for initialization of resNet in [DTC+2023], but they do discuss initialization for MLC (left column on page 5456). It is not clear how to compare these forms of initialization and their significance for the final ACC numbers. Do you have an advice?
>
> We would be happy to discuss  [DTC+2023] or other relevant SOTA if we missed something very recent. However, please note that our paper is more focused on studying conceptual properties of collision cross entropy (where Tables 1,2,3 are more useful) as a general loss that can be useful in many applications.
>
> **Question 1**\
> u is a uniform distribution.
>
> **Question 2**\
> This is what latex gives us, we do not use anything special in (16) :(
>
> **Question 3**\
> Thanks for catching this, will fix.

---

> > ### Comment · Reviewer_pteP · 2024-08-13
> > **Response**
> >
> > I would like to thank the authors for their response.
> >
> > > We did not find the details for initialization of resNet in [DTC+2023], ... Do you have an advice?
> >
> > From Table 3 of [DTC+2023], I think they applied the MoCoV2 as the pre-trained model.
> >
> > I still find the contribution of this paper is limited: theoretically, it does not provide theoretical results on justifying why this new loss function; empirically, the performance is not better compared to previous work [DTC+2023]. For now I keep my original score.
> >
> > [CFG+2020] Xinlei Chen, Haoqi Fan, Ross Girshick, and Kaiming He. Improved baselines with momentum contrastive learning. Mar. 2020.

---

> ### Author Response · Authors · 2024-08-14
>
> > it does not provide theoretical results on justifying why this new loss function
>
> This statement demonstrates the lack of good faith in providing an objective review and we would like to point this out to the Area Chair. This is not acceptable. The statement that we do not have any theories motivating CCE contradicts an obvious observation that most of our paper is a theoretical/numerical/conceptual motivation for CCE discussing limitations of the Shannon's cross-entropy. You did not even acknowledge our earlier replies to Theory-1 and Theory-2 as if they do not exist. Have you missed them?
>
> > SOTA experiments in [DTC+2023] paper
>
> The main contribution of our paper is a new (for network training) version of cross entropy (CCE) as an alternative to standard Shannon's cross-entropy. Our validation of CCE wrt standard entropy was focused on comparison with other methods using entropies (including SCAN and others in our Table 4). These and other results (including our Fig 2) empirically validate the advantages of the proposed CCE. Do you see any specific reasons to disagree?
>
> Our paper is not focused on SOTA. But, it does not mean that we do not care. We thank the reviewer for pointing out some references, e.g.  [DTC], that we can happily reference, discuss, and point out their better numbers. Please note that achieving SOTA in clustering requires many tricks. For example, there are three versions of SCAN (see their original paper) and the results get better and better as more tricks are added. Their pure clustering part using entropy losses achieves only 81.8 ACC on CIFAR10, but with other tricks it goes up to 87.4. The result in DTC paper (MLC) achieves 92.2 ACC using MoCo pertaining (thanks for clarifying this). Strangely, they do not show their MLC result for SimCLR pretraining in the first block in their Table 4. This result would have been the most relevant to compare with our number (83.3) since we use SimCLR pretraining. Also, note that SCAN-SimCLR number 87.6 reported in DTC paper is higher than the number we compare with for SCAN in our paper (81.8). We selected the SCAN result without any extra tricks for a fair comparison.
>
> In short, we do not believe that it is easy to make any solid conclusions about SOTA based on Table 4 in DTC paper, but we will gladly add this reference and discuss the challenges in establishing the truth in the state of the art.

---

### Decision · Program_Chairs · 2024-09-25

**Decision:**

Reject

**Comment:**

The manuscript proposes collision cross-entropy, a generalization of cross-entropy, for tasks with soft labels, such as self-labeled clustering.

The reviewers acknowledged that the introduction of collision cross-entropy is novel, well-motivated, and well-presented, they had reservations preventing a recommendation of acceptance. Most notably, the experimental setup is not sufficiently convincing, since the methods for comparison are not SOTA, and that fair comparisons are difficult due to differences in pre-processing (which is, admittedly, a common problem). I further concur with Reviewer VSk7 in the position that an ablation study is important that tries to disentangle the effects of the EM-type algorithm and the new cost function. Reviewer Puk1 in addition suggested to analyze a larger class of entropy measures -- while this is certainly a promising idea, I agree with the authors that this would be out of scope of the present manuscript.